# Unraveling surface structures of gallium promoted transition metal catalysts in $CO_2$ hydrogenation

Si Woo Lee [1], Mauricio Lopez Luna[1], Nikolay Berdunov[1], Weiming Wan [1], Sebastian Kunze[1], Shamil Shaikhutdinov [1] & Beatriz Roldan Cuenya [1]

Gallium-containing alloys have recently been reported to hydrogenate $CO_2$ to methanol at ambient pressures. However, a full understanding of the Ga-promoted catalysts is still missing due to the lack of information about the surface structures formed under reaction conditions. Here, we employed near ambient pressure scanning tunneling microscopy and x-ray photoelectron spectroscopy to monitor the evolution of well-defined Cu-Ga surfaces during $CO_2$ hydrogenation. We show the formation of two-dimensional Ga(III) oxide islands embedded into the Cu surface in the reaction atmosphere. The islands are a few atomic layers in thickness and considerably differ from bulk $Ga_2O_3$ polymorphs. Such a complex structure, which could not be determined with conventional characterization methods on powder catalysts, should be used for elucidating the reaction mechanism on the Ga-promoted metal catalysts.

Severe environmental issues, caused by the combustion of fossil fuels and waste resulting in the continuously increasing level of atmospheric $CO_2$[1,2], force the catalysis community to devote enormous attention to conversion of $CO_2$ into useful chemicals such as methanol[3–5]. Significant progress has been made by developing $Cu/ZnO/Al_2O_3$ catalysts for $CO_2$ hydrogenation at moderate temperatures 473–573 K and high pressures, i.e. 50–100 bar[6–9]. This methanol synthesis does not only bring along safety risks and a high energy consumption, but also limits the $CO_2$ concentration in the gas feed to about 10% in order to maintain high selectivity[10]. Therefore, a new class of catalyst materials for low pressure methanol synthesis is highly desirable. Reducing of the reaction pressure is also important for the future development of small-scale devices using solar-generated hydrogen at ambient pressures.

Recently, it has been discovered that Ni-Ga intermetallic catalysts show good catalytic performance even at atmospheric pressures, with similar or higher methanol yield than the $Cu/ZnO/Al_2O_3$ methanol synthesis catalyst under the same reaction conditions[11,12]. This finding triggered studies aimed to comprehend the Ga promoter effect on Ni[10,13,14] and other transition metal (Cu, Pd) catalysts[15–18]. However, the role of Ga in these catalysts is still poorly understood, primarily because of the lack of information about the *surface* structures of the catalysts. In this respect, studies using surface-sensitive techniques applied to well-defined model systems under reaction conditions can provide key information to elucidate the dynamic nature of active sites, reaction intermediates, and reaction mechanism, and hence provide the basis for the rational design of Ga-promoted catalysts.

In this work, we took advantage of Near Ambient Pressure X-ray Photoelectron Spectroscopy (NAP-XPS) and Scanning Tunneling Microscopy (NAP-STM)[19–26], to monitor in situ the structural and chemical evolution of the Ga-Cu bimetallic surfaces in the $CO_2$ hydrogenation reaction. We observed temperature- and pressure-dependent de-alloying of the bimetallic surface resulting in Ga-oxide islands embedded into the Cu surface. Although the oxide phase showed a stoichiometry close to $Ga_2O_3$, i.e., the most stable Ga-oxide, it actually forms an ultrathin layer having no bulk counterpart. Therefore, the $GaO_x/Cu$ interface formed under reaction conditions may expose catalytically active sites never considered for this reaction before. Such information would be impossible to obtain using bulk-sensitive techniques primarily employed for characterization of powder catalysts.

[1]Department of Interface Science, Fritz Haber Institute of the Max Planck Society, 14195 Berlin, Germany. ✉e-mail: shaikhutdinov@fhi-berlin.mpg.de

## Results

### Preparation of the Ga-Cu model catalysts

The Ga-Cu bimetallic surfaces were prepared by physical vapor deposition of Ga onto the clean Cu surface in ultra-high vacuum (UHV) (see Methods and Supplementary Fig. 1). We focus on the Cu(111) surface as the most representative for octahedrally-shaped metal nanoparticles. The amount of Ga was varied by the deposition time and controlled by XPS using the ratio of the Ga $2p$ and Cu $2p$ signals (Supplementary Fig. 2). The clean Cu(111) surface exposed atomically flat wide terraces separated by monoatomic steps (Fig. 1a). Deposition of Ga at room temperature at sub-monolayer coverages resulted in randomly distributed monolayer islands of a poorly defined shape (Fig. 1b), indicating the formation of diffusion-limited metastable structures at these temperatures. Upon heating in UHV to elevated temperatures the islands disappear, ultimately resulting in atomically flat terraces at 600 K (Fig. 1c). However, low energy electron diffraction (LEED) patterns of the annealed surfaces revealed the ($\sqrt{3} \times \sqrt{3}$) $R30°$-Cu(111) surface (or c(2×2)- in another notation) (Supplementary Fig. 3). In full agreement with LEED, high-resolution STM images showed a close-packed lattice of protrusions forming the unit cell of ~4.4 Å, which is rotated by 30° with respect to the Cu(111) lattice (inset in Fig. 1c). Interestingly, the formation of the c(2×2)Ga-Cu(111) surface upon vacuum annealing does not depend on how much Ga has been deposited (Supplementary Fig. 4). The Ga $2p$: Cu $2p$ signal ratios measured on the annealed samples are almost the same, irrespective of the initial Ga coverage in the "as deposited" samples varied between 0.5 and 3 monolayers (ML) (Supplementary Fig. 2a, b). The fact that the Ga:Cu ratio in the annealed samples considerably increases at grazing electron emission (Supplementary Fig. 2c) suggests that Ga is mostly located in the surface layers and forms a surface alloy. Since the annealing temperature (600 K) is too low for Ga desorption into the vacuum to occur (the Ga vapor pressure at 600 K is only $10^{-15}$ mbar[27]), we conclude that the Ga atoms not involved in the surface alloying readily diffuse into the Cu crystal up to a distance larger than the escape depth of the Ga $2p$ photoelectrons (i.e., 1–2 nm). Such a migration of the surface Ga atoms into the bulk may have an impact on the Ga distribution in bimetallic nanoparticles formed in the Ga-promoted catalysts, and even underlie the particle size effect on their reactivity.

### Real-time observation of surface morphology under reaction conditions

The well-ordered Ga-Cu(111) surfaces were studied in-situ with NAP-STM. Figure 2a displays an STM image recorded in the $CO_2 + H_2$ (1:3) reaction mixture at $10^{-2}$ mbar at room temperature. The large-scale morphology is unchanged under these conditions. Moreover, high-resolution STM images (Fig. 2d) continue showing the c(2×2) structure

as on the pristine surface in UHV. When the pressure increased to 1 mbar, numerous dark spots start to appear randomly on the surface (Fig. 2b), which were not observed on pure Cu(111) in the same reaction mixture (Supplementary Fig. 5). Meanwhile, the areas between the spots now show a (1×1) periodicity as on the clean Cu(111) surface (Fig. 2e). The spot density and their size increase when the sample is heated to 373 K (Fig. 2c), while the terraces still show the (1×1) structure (Fig. 2f). Such morphology basically remains at 473 K (Fig. 2g).

Post-characterization of the 473 K-treated sample by XPS in the same UHV setup revealed Ga in a fully oxidized state (Fig. 2h). The binding energy (BE) of the Ga $2p_{3/2}$ level (1117.6 eV) is shifted by 1.5 eV from 1116.1 eV in the pristine sample. Meanwhile, Cu is only found in the metallic state (Cu LMM Auger lines are shown in Fig. 2h; for Cu $2p$ spectra, see Supplementary Fig. 6). The O $1s$ core level is more complex, as it shows not only lattice oxygen ($O^{2-}$) in Ga-oxide (at 530.2 eV), but also other states at 531.4 and 532.2 eV associated with $CO_2$ adsorption (see more below). To complete the STM and XPS correlation, we also present the spectra obtained on an identical but newly prepared sample after a NAP-STM study at 1 mbar in the same reaction mixture at 300 K (see (ii) in Fig. 2h). The combined STM and XPS results show that the Ga-Cu surface alloy undergoes de-alloying in the $CO_2 + H_2$ atmosphere, resulting in Ga-oxide islands surrounded by the Cu(111)-(1×1) surface. Apparently, the Ga-Cu surface is oxidized via $CO_2$ dissociation, providing O ad-atoms to react with Ga.

### Chemical transformation of the Ga-Cu alloy surface during the reaction

To monitor the chemical state of our model catalysts in situ, we carried out NAP-XPS measurements. Figure 3a shows consecutive spectra recorded on the c(2×2)Ga-Cu(111) surface in the reaction mixture of 1 mbar at 300 K. It is clear that Ga slowly oxidizes in the reaction atmosphere: the Ga $2p_{3/2}$ signal at 1116.1 eV attenuates, while the 1117.6 eV signal gains in intensity. Also the O $1s$ signal grows in time, showing two components from the onset, i.e., at 530.2 and 531.4 eV, as obtained by spectral deconvolution. (The signal at 536.9 eV originates from $CO_2$ in the gas phase.) Using NAP-XPS data for pure Cu(111) under the same conditions for comparison (Supplementary Fig. 7), we assign the 530.2 eV signal to $O^{2-}$ ions in the Ga-oxide and the 531.4 eV signal to $CO_2$ adsorption on Cu(111) via carboxylate $(CO_2^{\delta-})$[28–30]. It is interesting that the $CO_2$-related signals on pure Cu(111) appear immediately upon gas exposure and stay constant with time (Supplementary Fig. 7), whereas the adsorption processes are much slower on the Ga-Cu surface and proceed along with the formation of the Ga-oxide (Fig. 3a). This finding suggests that the Ga atoms in the surface layer actually weaken the interaction with $CO_2$. Since $CO_2$ dissociation probability even on the clean Cu(111) surface is very low[29], we cannot exclude that Ga is oxidized by traces of $O_2$ in the reaction mixture (see also ref. 29).

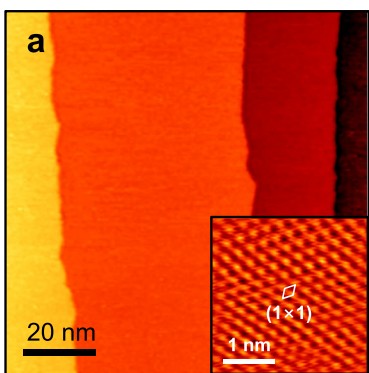
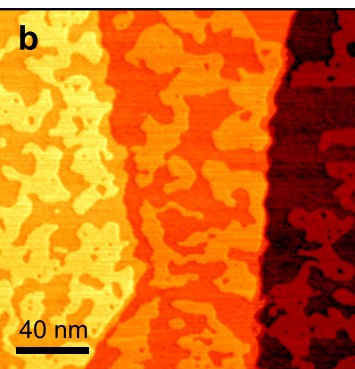
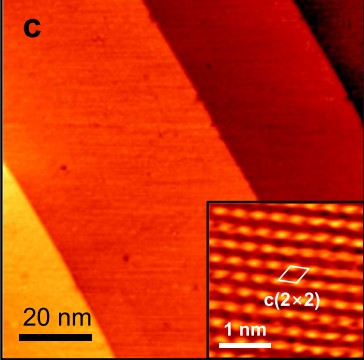

**Fig. 1 | Preparation of the Ga-Cu surface alloy.** Room-temperature STM images obtained in UHV: **a** the clean Cu(111) surface (tunneling parameters: sample bias 0.5 V, current 1 nA; 0.2 V, 4 nA (inset)); **b** after 0.5 ML Ga deposition at 300 K (1.4 V, 0.2 nA); **c** after subsequent UHV annealing at 600 K for 15 min (0.3 V, 1 nA; 0.1 V, 7 nA (inset)). The surface unit cells are indicated in the atomically-resolved images shown in the insets.

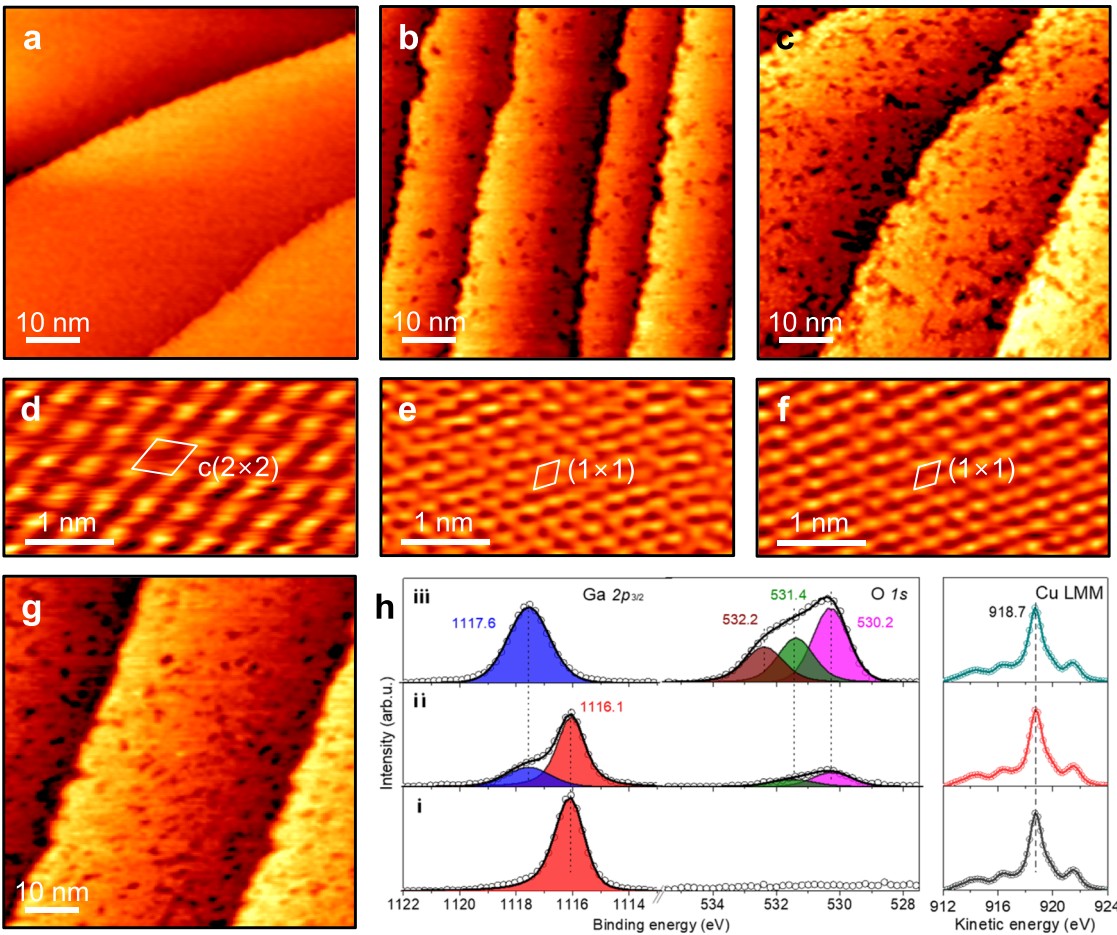

**Fig. 2 | Structural evolution of the Ga(√3×√3)R30°-Cu(111) surface under reaction conditions.** NAP-STM images obtained in $CO_2 + H_2$ (1:3) reaction mixture: **a** $10^{-2}$ mbar at 300 K (0.5 V, 0.3 nA); **b** 1 mbar at 300 K (0.4 V, 1 nA), **c** 1 mbar at 373 K (0.3 V, 1 nA), and **g** 1 mbar at 473 K (0.3 V, 1 nA). **d–f** The corresponding high-resolution NAP-STM images, all measured at 0.1 V and 6 nA, are shown below (**a–c**), respectively. **h** XPS spectra of the Ga 2p, O 1s and Cu LMM Auger lines measured in UHV before (i) and after the NAP-STM measurements in 1 mbar at 300 K (ii) and 473 K (iii).

Accordingly, surface de-alloying re-opens the clean Cu(111) surface that starts to adsorb $CO_2$ via carboxylate.

More complex spectra were obtained with increasing temperature (Fig. 3b). At 373 K, a new state at 532.2 eV appears in the O 1s region, whereas the 531.4 eV signal assigned to $CO_2^{\delta-}$ decreases and ultimately disappears at 473 K in the same manner as on pure Cu(111), see Supplementary Fig. 7. Cu remains metallic in all these experiments (Supplementary Fig. 8)[31]. Therefore, all O species observed at high temperatures must be associated with Ga. Note however, that the Ga-oxide phase is partially reduced at increasing temperature (see the Ga 2p spectra in Fig. 3), apparently in the same way as the ZnO phase in the Cu-Zn catalysts previously studied by *operando* X-ray absorption spectroscopy and NAP-XPS[32]. The reduction also manifests itself via the reduced intensity of the $O^{2-}$ signal at 530.2 eV that made possible the deconvolution of another state centered at 531.0 eV, which dominates the spectra at high temperatures. Basically, the same spectral evolution was observed for all c(2×2)Ga-Cu(111) samples studied, irrespectively of the initial Ga coverage (Supplementary Figs. 9–14).

The O 1s states at 532.2 and 531.0 eV were only observed on the Ga/Cu(111) surfaces and must therefore be related to the formation of Ga-oxide upon exposure to the reaction atmosphere. Analysis of the C 1s spectra obtained in these experiments (Supplementary Fig. 15) could not shed more light on the nature of these O states. To the best of our knowledge, there are no XPS studies addressing $CO_2$ and $H_2$ adsorption on clean and well-defined Ga-oxide surfaces to compare with. Based on infrared spectroscopy studies, $Ga_2O_3$ and Ga-promoted

metal catalysts in $CO_2$ and $H_2$ atmospheres may form carbonates ($CO_3^{2-}$), bicarbonates ($HCO_3^-$), formate ($HCOO^-$), hydroxyls (−OH), and methoxy ($CH_3O-$) species[15,17,18,33]. Therefore, we carried out additional XPS measurements on the benchmark samples as described in Supplementary Note 1. The 532.2 eV signal bears close similarity to that observed in previous studies on Cu single crystal surfaces modified with ZnO and $CeO_2$ and assigned to formate[34–36]. Their formation would suggest $H_2$ dissociation on the Cu(111) surface surrounding the Ga-oxide islands, and subsequent H spillover onto the oxide surface. The hydrogen dissociation is also a precondition for the Ga-oxide reduction to metallic Ga (Fig. 3b). The origin of the 531.0 eV peak, which dominates the O 1s spectra at high temperatures (Fig. 3b), remains unclear (Supplementary Note 2).

In order to evaluate whether the Ga-Cu surface undergoes the same structural/chemical evolution when exposed to more catalytically relevant pressures, our model catalysts were exposed to the $CO_2 + H_2$ (1:3) reaction mixture at 5 bar (total) in a high-pressure cell (see details in Supplementary Materials) and transferred back to the XPS/STM chamber without exposure to air. The obtained XPS spectra (Supplementary Fig. 18, Supplementary Note 3) look virtually identical to those measured after reaction at 1 mbar, suggesting that the reaction-induced surface transformations are already well captured in the mbar range. We note however, that the ex situ XPS spectra considerably differ from the in situ NAP-XPS spectra (Fig. 3b) recorded in the reaction atmosphere. In particular, Ga was found fully oxidized in the ex situ measurements, whereas the NAP-XPS spectra revealed Ga

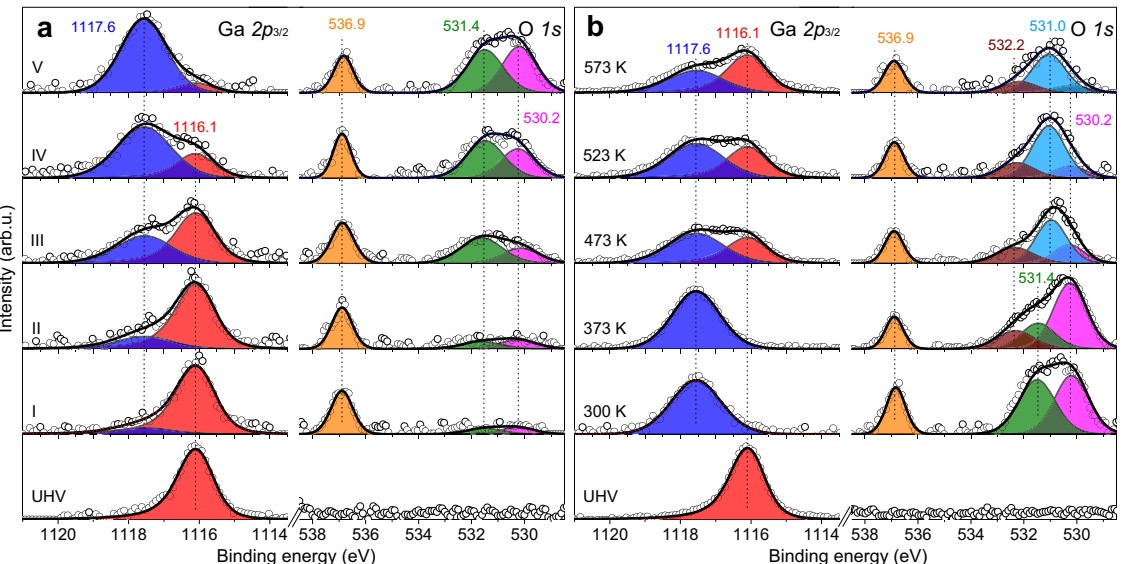

**Fig. 3 | Chemical evolution of the Ga(√3×√3)R30°-Cu(111) surface under reaction conditions.** NAP-XPS spectra of the Ga 2p and O 1s states measured on the Cu-Ga alloy surface in 1 mbar of the $CO_2 + H_2$ (1:3) reaction mixture. **a** Consecutive spectra (from bottom to top, the acquisition time is 27 min per spectra (i) – (v)) measured at 300 K; **b** "Steady state" spectra measured at the reaction temperature of 300–573 K, as indicated. All spectra in (**b**) are an average of five consecutive spectra, except for 300 K, for which the last spectrum (v) from the (**a**) is reproduced.

partially reduced at high reaction temperatures. Furthermore, the ex situ O 1s spectra only showed 532.2 and 531.4 peaks, whereas the 531.0 eV state, clearly observed in the NAP-XPS spectra (Fig. 3), could not be identified here as it is obscured by the strong signal of lattice oxygen at 530.2 eV. Therefore, the ex situ XPS measurements of the reacted catalysts, even after precluding air exposure, cannot provide full information about the surface composition of the Ga-Cu catalysts under reaction conditions, thus highlighting the important role of in situ investigations for mechanistic understanding of surface reactions. In addition, Supplementary Fig. 19a, b displays STM images of the Cu and Ga-Cu surfaces after reaction at 5 bar. The original Cu(111) terraces are still recognizable on both samples, suggesting the absence of a surface reconstruction involving severe mass transport even under high pressure conditions. Nonetheless, the post-reacted Ga-Cu surface exhibited a considerably higher corrugation amplitude than that of the Cu counterpart, as inferred from the topography profiles.

**Ambient induced formation of an ultrathin Ga oxide layer on Cu**
Further analysis of the NAP-XPS spectra (Fig. 3) at high reaction temperatures showed that the degree of Ga-oxide reduction measured via the Ga(metal)/Ga(total) signal ratio is considerably smaller than that deduced from the normalized intensity of the corresponding lattice oxygen peak at 530.2 eV (Supplementary Fig. 20). Since the Ga-oxide phase does not decompose at these temperatures if heated in vacuum in the blank experiments, this finding suggests a large fraction of the O atoms being involved in the formation of ad-species and giving rise to the signals at higher BEs at the expense of the lattice oxygen signal. The adsorbate-induced reduction of the lattice oxygen peak is small in the case of bulky oxide samples, but becomes substantial in the case of ultrathin oxide films. This was shown, for example, for $CeO_2$ and ZnO "monolayer" ad-islands on Cu(111), with the island heights being directly measured by STM[37,38]. Therefore, we can conclude that our Ga-oxide islands observed by STM are ultrathin in nature, i.e., consist of only a few layers.

To get more insight into the atomic structure of the Ga-oxide formed under $CO_2$ hydrogenation conditions, we prepared the $GaO_x$/Cu(111) surface by annealing the c(2×2)Ga-Cu(111) surface in $10^{-6}$ mbar of $O_2$ at 600 K. First, the XPS spectra presented in Supplementary Fig. 21 support the assignment of the Ga 2p peak at 1176.6 eV and of the

O 1s peak at 530.2 eV to gallium oxide. The molar Ga:O ratios determined by XPS for Ga-oxides formed in $10^{-6}$ mbar of $O_2$ and in 1 mbar of $CO_2 + H_2$ are 0.65 (±0.03) and 0.68 (±0.05), respectively, suggesting a $Ga_2O_3$ stoichiometry in both cases. Moreover, STM images in Fig. 4a, b revealed dark spots randomly distributed on the Cu(111) surface, which look similar to those observed in the NAP-STM experiments in the $CO_2 + H_2$ reaction atmosphere (Fig. 2). Importantly, their appearance is independent of the bias voltage and polarity used (between −1 V and +1 V), indicating that the $GaO_x$ islands are, in fact, not ad-islands, but embedded into the topmost Cu(111) layers, as schematically shown in Fig. 4d. If the effect would be purely electronic, an image contrast would depend on the tunneling parameters[39]. It should be noted that the $GaO_x$ islands formed on the Cu(001) substrate studied by us previously[40] also appeared as dark spots. Therefore, Ga-oxide islands on Cu seem to form embedded structures independently of the Cu surface orientation.

To the best of our knowledge, the formation of oxide islands embedded into a "host" metal surface has only been documented for the NiO(100)(1×1)-Ag(100) surface, where the NiO islands were no more than two layers in thickness[41]. In general, the morphology of oxide overlayers on metals is a result of a delicate balance between the surface and interface energies, which in turn depend on the oxide and the metal involved, as well as the lattice mismatch. In the case of embedded structures the interfacial energy at the island edge may also contribute to the final structure of the nano-sized islands observed here. Unfortunately, on such small islands we could not get STM images with truly atomic resolution. Nonetheless, we imaged atomic rows with ~ 11 Å spacing running in the three azimuthal directions rotated by 120° (Fig. 4b) and parallel to the crystallographic orientations on the Cu(111) surface. In attempts to better visualize the atomic structure of the Ga-oxide surface with STM, we prepared samples with larger $GaO_x$ patches by varying the Ga coverage and oxidation temperature. The corresponding STM images (Fig. 4c) revealed a close-packed lattice of protrusions with a ~ 3.1 Å periodicity which runs in the same crystallographic orientation as the Cu(111) underneath. This implies certain epitaxial relationship of the Ga-oxide overlayer and the Cu(111) support. Moreover, the surface showed a complex "zigzag"-like superstructure. Similar structures have been previously observed for "monolayer" $TiO_x$ films on Pt(111) and Pd(111)[42–44] and $AlO_x$ films on

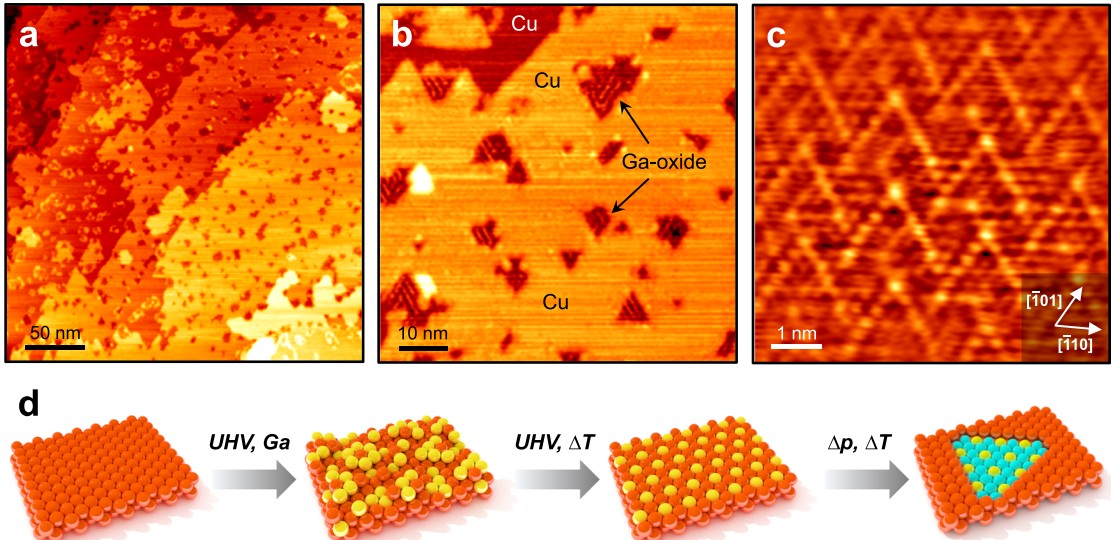

**Fig. 4 | Morphology of the Ga-Cu(111) surface oxidized in O₂. a, b** STM images of the Ga(√3×√3)*R*30°-Cu(111) surface oxidized in $10^{-6}$ mbar of $O_2$ at 600 K (0.3 V, 0.2 nA). **c** High-resolution STM image (−0.3 V, 0.9 nA) of the extended Ga-oxide surface showing a close-packed lattice of atomic protrusions with a 3.1 Å periodicity, in turn forming a zigzag-like superstructure. The crystallographic orientation of the Cu(111) surface is shown. **d** Schematic representation of the structural evolution of the Cu(111) surface after Ga deposition in vacuum, subsequent annealing in vacuum, and exposure to the reaction gases ($CO_2$ + $H_2$ or $O_2$). Cu – red, Ga – yellow, and O – cyan.

NiAl(110)[45,46]. Therefore, in full agreement with the XPS-derived conclusion about the very small thickness of the $GaO_x$ islands, STM data provide further evidence that the Ga-oxide islands are, indeed, ultrathin.

In summary, our results point to a complex *surface* structure of the Ga-promoted Cu catalysts under reaction conditions. Such information is extremely difficult to obtain using primarily bulk-sensitive techniques commonly employed for characterization of powder catalysts. Using NAP-STM and NAP-XPS methods operated at near ambient pressures and catalytically relevant temperatures, we showed that an ultrathin Ga-oxide layer is always present at the metal surface under $CO_2$ hydrogenation reaction conditions.

The promotional effect of metals like Ga which are prone to oxidation is often discussed within structure models, where *a bulk* $Ga_2O_3$ oxide is placed *on top* of the metal surface and the corresponding reaction mechanism involves spillover of intermediate species at the interface, often not identified experimentally. In contrast, our study clearly demonstrates that: (i) Ga-oxide is embedded *into* the metal surface as small islands, albeit their lateral size may depend on the Ga coverage (loading); and (ii) the Ga-oxide islands are ultrathin, most likely of "monolayer" thickness. Ultrathin, in essence two-dimensional, oxide films are very different from bulky counterparts in terms of both, structure and reactivity[47–51]. Even though the Ga-oxide phase formed on Cu(111) shares a $Ga_2O_3$ stoichiometry, it structurally differs from any known bulk phases of $Ga_2O_3$. At least, none of those could expose a close-packed arrangement either of Ga or O atoms imaged in STM as protrusions with a 3.1 Å periodicity (Fig. 4c) and simultaneously be below 1 nm in the direction normal to the surface[52,53]. Another aspect is that the surface lattice constant measured on the Ga-oxide layer (~3.1 Å) is much larger than that of Cu(111) (2.56 Å), implying considerable strain deformation. The latter may be the reason for the row-like structure observed within Ga oxide nano-islands in Fig. 4b. Moreover, the formation of ultrathin Ga-oxide films is anticipated also for the surfaces of intermetallic compounds. For example, this has been demonstrated for CoGa(100) and (110) surfaces[54,55].

Our experimental results shed light on the complex surface structure of Ga-containing catalytic systems, which is only possible to obtain using state-of-the-art experimental techniques under reaction conditions. Only by establishing the atomic structure of the Ga-oxide layer(s) and its interface to the transition metal under working conditions can one bring insight into the reaction mechanism of this methanol synthesis catalyst.

## Methods

The experiments were carried out in two multi-chamber UHV setups (from Specs). The first setup consists of a preparation chamber; an analytical chamber equipped with XPS and LEED; and a NAP-STM chamber. For sample treatments at elevated pressures up to 5 bar, a high-pressure cell (Specs HPC20) is attached to the preparation chamber via a gate valve. The second setup consists of a similarly equipped preparation chamber and a NAP-XPS chamber.

### Surface preparation

A Cu(111) single crystal (from MaTeck GmbH, 99.99% purity, 9 mm in diameter and 2 mm in thickness, a surface miscut below 0.5°) was clamped to a stainless steel sample holder having a hole (~9 mm diameter) to heat the crystal from the backside via electron bombardment using a thoriated tungsten filament. The crystal temperature was measured with a chromel-alumel thermocouple mounted into the small hole at the edge of the crystal. The clean Cu(111) surface was prepared using several cycles of $Ar^+$ ion sputtering (sample current 15 μA) at 300 K for 30 min and UHV annealing at 900 K for 5 min, until no contamination was detected by XPS, and STM images revealed atomically flat wide terraces.

Gallium (Sigma Aldrich, 99.9995%) was deposited onto the clean Cu(111) surface at room temperature using an electron-beam assisted evaporator (Focus EFM3) from a boron nitride crucible placed in a Mo liner. The Ga deposition flux was kept constant (monitored by the integrated flux monitor, 3 nA in this case), and the amount of deposited Ga was varied by the deposition time, typically 5–30 min. The Ga coverage was controlled via the Ga $2p_{3/2}$: Cu $2p_{3/2}$ XPS signals ratio measured in UHV at 300 K. For this study, all samples were annealed in UHV at 600 K for 15 min to ensure homogenous intermixing at the surface and to exclude pure thermal effects during measurements at elevated temperatures.

In all experiments using the reaction mixture, we used the ratio $CO_2: H_2 = 1: 3$ that reflects a stoichiometric equation for the methanol synthesis reaction $(CO_2 + 3H_2 \rightarrow CH_3OH + H_2O)$ and has been commonly used in related prior studies.

### NAP-STM measurements

NAP-STM images at variable temperature were obtained with an Aarhus STM 150 NAP instrument (Specs). The integrated reaction cell (ca 15 ml) is separated from UHV using a Viton O-ring. The reaction $CO_2 + H_2$ (1:3 ratio) mixture was introduced into the cell using a precision leak valve connected to $CO_2$ and $H_2$ (both of 5 N purity) aluminum gas cylinders. The pressure in the reaction cell was monitored with a full-range gauge (Pfeiffer Vacuum). The sample can be heated up to 473 K using a halogen lamp behind the sample. XPS spectra of the samples after the NAP-STM experiments showed no contamination. STM images were recorded in the constant current mode with an electrochemically etched tungsten tip which was preconditioned by $Ar^+$ ion sputtering and subsequent scanning over the clean Au(111) surface.

### NAP-XPS measurements

NAP-XPS measurements were performed using a monochromatic Al $K_\alpha$ x-ray source (hv = 1486.6 eV) and differentially pumped hemispherical analyzer (Phoibos 150 NAP, Specs). High-purity $CO_2$ and $H_2$ gas cylinders were connected to a gas manifold line, which was pumped down to $10^{-7}$ mbar and baked out at 393 K before introducing the reaction gases. $CO_2$ and $H_2$ were mixed with the 1:3 molar ratio and controlled by high-precision flow-meters (Bronkhorst). The samples were heated from the backside using a halogen lamp. In both XPS and NAP-XPS setups, the spectra were recorded at normal emission and a pass energy of 20 eV. Binding energies were calibrated using the Fermi edge of Cu(111). The spectra were analyzed with a commercial CasaXPS software (version 2.3.19). The spectrum background was subtracted using a Shirley-type baseline. The spectra of Cu 2p and Ga 2p metal states were fitted with asymmetric line-shape, whereas the O 1s spectra and Ga 2p in the oxidized state—with a mixed Gaussian and Lorentzian function.

## Data availability

The authors declare that the data supporting the findings of this study are available within the paper and its supplementary information files, and also available from the corresponding author upon reasonable request.

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

## Acknowledgements

The work was supported by the European Research Council (ERC-725915, OPERANDOCAT) and the Deutsche Forschungsgemeinschaft (DFG) under Germany´s Excellence Strategy – EXC 2008 – 390540038 – UniSysCat, and also by the German Federal Ministry of Education and Research (BMBF) under Grant No. 03EW0015B (CatLab). S.W.L. and W.W. thank Alexander von Humboldt Foundation for the postdoctoral fellowships.

## Author contributions

S.W.L. performed the majority of experiments with the assistance of W.W., M.L.L., N.B., and S.K. S.S. and B.R.C. conceptualized the research. S.S. wrote the manuscript with the feedback of all co-authors.

## Funding

## Competing interests
The authors declare no competing interests.
