## [Peer Review File · Nature Communications]

Unraveling surface structures of Ga-promoted transition metal catalysts in CO₂ hydrogenationREVIEWER COMMENTS

Reviewer #1 (Remarks to the Author):

The manuscript by Lee et al on the CO₂ hydrogenation studies on Ga-promoted Cu(111) surfaces is a very interesting AP-XPS and AP-STM study and it is worth publishing in Nature Comm. First and foremost, there are only a handful of such studies that combine these techniques in the literature and there is definitely a need for more. This study is an excellent example of how powerful this approach is. The experiments and data processing are done thoroughly, the results are sound, and the discussion is interesting to read. I only have two minor remarks.

1- I would like to bring the authors' attention to the following papers:

Salmeron group has studied ultra-thin (monolayer) oxides with AP-XPS and AP-STM in the recent years and has claimed that they might indeed be the active regions.

Nature Catalysis 2019, 2, 78–85 (this paper is actually cited, but not in the mentioned context)

J. Am. Chem. Soc. 2020, 142, 8312–8322

In both cases, CO oxidation was studied.

There is also a recent study, where the effect of dosing order of CO₂ and H₂ on Cu was investigated with AP-XPS and NEXAFS.

J. Am. Chem. Soc. 2023, 145, 12, 6730–6740

If the authors think these three papers are relevant, they should cite them wherever convenient.

2- In terms of atomic / sub-nm structure, the manuscript is perfect. I do not have any questions to raise. I understand that this is the main purpose of this work (from the title, abstract, conclusions – this is what I understand). However, in the discussion, there is also a claim about products/intermediates solely based on the O 1s data. This is a slippery slope because the authors do not have C 1s data to back up their claim of methoxy or formate formation. I am assuming this is because they do not get a good SNR in the C 1s data due to Al K α emission (both low photoionization cross-section, and low surface-sensitivity). Moreover, they also do not use other techniques that can help them distinguish between these surface species with a better accuracy, such as infrared spectroscopy. I highly suggest the authors to tone down this claim because it might turn out to be incorrect in future studies. They do not have to fully remove this claim, but use more careful wording.

Finally, I would like to congratulate the authors for this fantastic work.

Reviewer #2 (Remarks to the Author):

This is a very interesting study, but there are a series of key points that need clarification and it is not clear to me that this paper needs to be published in Nature Communications or in a more focused journal in catalysis.

1) Lines 39-40: What do the authors mean by "excellent performance"? Why is the Ni-Ga catalyst superior with respect

to other catalysts reported in the literature for CO₂ hydrogenation?

2) Line 50: What do the authors mean by "realistic pressure conditions"? CO₂ hydrogenation processes are typically done at high pressures. Can NAP-XPS and NAP-STM be operated under high pressures of CO₂ or H₂?

3) Lines 60-80: What is the link between the Ga-Cu(111) model and a real GaCu catalyst? I agree that is very important to study the behavior of Ga-Cu(111), but how far can one extrapolate the results obtained for this surface?

4) Lines 60-80: What is the miscibility of Ga and Cu? In the systems show in Figure 1, are we seeing a surface alloy or a bulk alloy?

5) Figure 2, it would be nice to see sequential STM images for

a pristine Ga-Cu(111) surface, a surface exposed to CO₂, and a surface exposed to CO₂ + H₂. Same thing for the NAP-XPS results. It is important to establish what CO₂ alone can do to the Ga-Cu(111) surface.

6) Lines 142-158: The best way to study the possible formation and disappearance of the mentioned hydrocarbon intermediates is by measuring the C 1s XPS or C K-edge NEXAFS.

7) Lines 295-306: Why the authors did not measure C 1s XPS spectra? Neither Ga nor Cu have XPS features in the 280-290 eV region.

Reviewer #3 (Remarks to the Author):

Transition-metal based catalysts, for instance, ZnO-Cu, CeO_x-Cu, have been widely investigated in the field of the heterogeneous catalysis towards CO₂ reduction reaction (CO₂RR). Especially, Gallium-containing alloys are recently demonstrated to be able to hydrogenate CO₂ to methanol at mild conditions. In this context, this work unravels surface structures of the Ga-promoted Cu catalyst during CO₂ hydrogenation, and demonstrates the formation of ultrathin Ga-oxide layer at the metal surface under CO₂ hydrogenation reaction conditions, while the Cu substrate is always kept metallic during reaction. In addition, authors also claim that the Ga-oxide islands are embedded into the copper substrate. Indeed, such kind of structure information at surface/interface is relatively difficult to obtain by using the conventional techniques as commonly utilized for characterization of powder catalysts.

Although the formation of ultrathin Ga₂O₃ layer embedded in the substrate has not been frequently observed before, it is generally recognized that the morphology of oxide overlayers on metals is a result of a delicate balance between the surface and interface energies. Furthermore, the atomic insight into the Ga-oxide layer is still missing in this work. There are numerous measurements including NAP-STM

and NAPXPS measurements presented in the supporting information, which indeed took considerable efforts to perform, however, the highlight of this work is not that appealing for readers. From both the weakness of novelty as similar reports found in literature, and the lack of fundamental mechanism investigation either in the reaction path/products selectivity or in the origin of metallic behavior of the Cu substrate during reactions. Thus, I come to the conclusion with regret that draft under its current version is not convincing enough to be published in Nature communications, but more suitable for the specialized journal, for example, Journal of catalysis or Journal of physical chemistry letters. Meanwhile, there are also a few concerns needed to be addressed:

1. The ratio of CO₂:H₂ was chosen to be 1:3 in this work, is there any concern about this? Has authors tried different partial ratio of pressure, say, 1:2 or 1:4 to see if any difference?

2. 'To complete the STM and XPS correlation, we also present the spectra obtained on an identical but newly prepared sample after a NAP-STM study at 1 mbar in the same reaction mixture at 300 K...' in page 4, related spectra is not shown or clearly mentioned in the draft.

3. Figure 2c is labelled two times by mistake in Figure 2.

4. The deconvolution of O1s spectra with two peaks at 531.0 and 531.4 eV (binding energy), is not convincing since the difference is such tiny, especially in the case of the spectrum at 300 K in Figure 3b where the fitting is not proper. Thus, some parts of discussion might be revised accordingly.

Reviewer 1

The manuscript by Lee et al on the CO₂ hydrogenation studies on Ga-promoted Cu(111) surfaces is a very interesting AP-XPS and AP-STM study and it is worth publishing in Nature Comm. First and foremost, there are only a handful of such studies that combine these techniques in the literature and there is definitely a need for more. This study is an excellent example of how powerful this approach is. The experiments and data processing are done thoroughly, the results are sound, and the discussion is interesting to read. I only have two minor remarks.

1-1. *I would like to bring the authors' attention to the following papers:*

Salmeron group has studied ultra-thin (monolayer) oxides with AP-XPS and AP-STM in the recent years and has claimed that they might indeed be the active regions.

Nature Catalysis 2019, 2, 78–85 (this paper is actually cited, but not in the mentioned context)

J. Am. Chem. Soc. 2020, 142, 8312–8322

In both cases, CO oxidation was studied.

There is also a recent study, where the effect of dosing order of CO₂ and H₂ on Cu was investigated with AP-XPS and NEXAFS.

J. Am. Chem. Soc. 2023, 145, 12, 6730–6740

If the authors think these three papers are relevant, they should cite them wherever convenient.

Authors' response: We thank the reviewer for the very positive feedback to our work. We are also grateful for drawing our attention to the above-mentioned papers. In the revised manuscript, we cite the Nat. Catal. (2019) and JACS (2020) papers highlighting the importance of ultra-thin oxide films in chemical reactions. We also included the JACS (2023) reference to emphasize that prior work found that the Cu surface is in metallic state under the CO₂ hydrogenation conditions, which is in agreement to our findings.

1-2. *In terms of atomic / sub-nm structure, the manuscript is perfect. I do not have any questions to raise. I understand that this is the main purpose of this work (from the title, abstract, conclusions – this is what I understand). However, in the discussion, there is also a claim about products/intermediates solely based on the O 1s data. This is a slippery slope because the authors do not have C 1s data to back up their claim of methoxy or formate formation. I am assuming this is because they do not get a good SNR in the C 1s data due to Al K α emission (both low photoionization cross-section, and low surface-sensitivity). Moreover, they also do not use other techniques that can help them distinguish between these surface species with a better accuracy, such as infrared spectroscopy. I highly suggest the authors to tone down this claim because it might turn out to be incorrect in future studies. They do not have to fully*

remove this claim, but use more careful wording.

Finally, I would like to congratulate the authors for this fantastic work.

Authors' response: We agree that precise identification of adsorbate species solely on the basis of O 1s data is quite difficult. Unfortunately, our C 1s spectra, which we include in the revision as Supplementary Figure 15 (see below), could not shed more light into the nature of the reaction intermediates. It should also be emphasized that even if additional experimental data such as IR data were available, and vibrational features corresponding to either methoxy or formate were detected, one could still not conclude if they are true intermediate species or simply spectators. One would in fact need theoretical calculations, which are presently not available. Following the reviewer suggestion, we have softened the above discussion in the main text and moved our mechanistic hypothesis to a Supplementary Note 1 in the SI, where we discuss the results of the benchmark experiments on CO₂ covered and hydroxylated surfaces:

Page 6: *“The O 1s states at 532.2 and 531.0 eV were only observed on the Ga/Cu(111) surfaces and must therefore be related to the formation of Ga-oxide upon exposure to the reaction atmosphere. Analysis of the C 1s spectra obtained in these experiments (Supplementary Fig. 15) could not shed more light on the nature of these O states. To the best of our knowledge, there are no XPS studies addressing CO₂ and H₂ adsorption on clean and well-defined Ga-oxide surfaces to compare with. Based on infrared spectroscopy studies, Ga₂O₃ and Ga-promoted metal catalysts in CO₂ and H₂ atmospheres may form carbonates (CO₃²⁻), bicarbonates (HCO₃⁻), formate (HCOO⁻), hydroxyls (-OH), and methoxy (CH₃O-) species^{15, 17, 18, 33}. Therefore, we carried out additional XPS measurements on the benchmark samples as described in Supplementary Note 1. The 532.2 eV signal bears close similarity to that observed in previous studies on Cu single crystal surfaces modified with ZnO and CeO₂ and assigned to formate^{34, 35, 36}. Their formation would suggest H₂ dissociation on the Cu(111) surface surrounding the Ga-oxide islands, and subsequent H spillover onto the oxide surface. The hydrogen dissociation is also a precondition for the Ga-oxide reduction to metallic Ga (Fig. 3b). The origin of the 531.0 eV peak, which dominates the O 1s spectra at high temperatures (Fig. 3b), remains unclear (Supplementary Note 2).”*

Supplementary Figure 15 | The C 1s region in the NAP-XPS spectra measured in 1 mbar of the $\text{CO}_2 + \text{H}_2$ (1:3) reaction mixture on pure Cu(111) (a) and Ga-Cu(111) (b) surfaces at different temperatures increased stepwise as indicated (from bottom to top). The corresponding O 1s spectra are displayed in Supplementary Figure 7 and Figure 3b in the main text, for Cu and Ga-Cu surfaces, respectively. The signal at 293.2 eV originates from CO_2 in the gas phase. The peak at 284.6 eV is assigned to adventitious carbon).⁽¹⁾

- (1) Deng X, Verdaguer A, Herranz T, Weis C, Bluhm H, Salmeron M. Surface Chemistry of Cu in the Presence of CO_2 and H_2O . *Langmuir* **24**, 9474-9478 (2008).

Supplementary Note 2: In the experiments described below in relation to Suppl. Fig. 17, the Ga-Cu sample treated in pure CO_2 was flashed to 500 K in UHV to desorb CO_2 -related adsorbates and subsequently exposed to 1 mbar of H_2 at 473 K for 30 min to hydroxylate the Ga oxide surface formed by CO_2 adsorption. A prominent signal at 531.8 eV, i.e., shifted by 1.6 eV with respect to oxygen in Ga-oxide (530.2 eV), falls in the range of the BE shifts (1.5 – 2.0 eV) reported for hydroxylated surfaces of the transition metal oxides⁽⁴⁾ and thus assigned to hydroxyls. Therefore, the 532.2 eV signal observed in $\text{CO}_2 + \text{H}_2$ can hardly be assigned to hydroxyls. In fact, this state bears close similarity to that observed on Cu surfaces modified with ZnO and CeO_2 , which was previously assigned to formate⁽⁵⁻⁷⁾. Hydroxyls can also not be the origin of the 531.0 eV state only appearing in the NAP experiments. This state also falls in the range observed for methoxy species obtained by direct adsorption of methanol on oxide surfaces^{8,9}. Nonetheless, it should be noted that the 531.0 eV state appears along with the Ga-oxide partial reduction (Fig. 3b) and may therefore be related to the O-deficient GaO_x

domains. Thus, a definitive assignment of the origin of these species cannot be made exclusively based on XPS data.

- (4) Dupin, J.-C., Gonbeau, D., Vinatier, P. & Levasseur, A. Systematic XPS studies of metal oxides, hydroxides and peroxides. *Phys. Chem. Chem. Phys.* **2**, 1319-1324 (2000).
- (5) Graciani, J. *et al.* Highly active copper-ceria and copper-ceria-titania catalysts for methanol synthesis from CO₂. *Science* **345**, 546-550 (2014).
- (6) Senanayake, S. D. *et al.* Hydrogenation of CO₂ to Methanol on CeO_x/Cu(111) and ZnO/Cu(111) Catalysts: Role of the Metal–Oxide Interface and Importance of Ce³⁺ Sites. *J. Phys. Chem. C* **120**, 1778 (2016).
- (7) Palomino, R. M. *et al.* Hydrogenation of CO₂ on ZnO/Cu(100) and ZnO/Cu(111) Catalysts: Role of Copper Structure and Metal–Oxide Interface in Methanol Synthesis. *J. Phys. Chem. B* **122**, 794 (2018).
- (8) Goodacre D, *et al.* Methanol Adsorption on Vanadium Oxide Surfaces Observed by Ambient Pressure X-ray Photoelectron Spectroscopy. *J. Phys. Chem. C* **125**, 23192 (2021).
- (9) Orozco I, *et al.* In Situ Studies of Methanol Decomposition Over Cu(111) and Cu₂O/Cu(111): Effects of Reactant Pressure, Surface Morphology, and Hot Spots of Active Sites. *J. Phys. Chem. C* **125**, 558-571 (2021).

Reviewer 2

This is a very interesting study, but there are a series of key points that need clarification and it is not clear to me that this paper needs to be published in Nature Communications or in a more focused journal in catalysis.

Authors' response: We thank the reviewer for the interest and positive feedback on our work. In our responses below, we address the technical questions raised, but first we want to highlight the implications of our findings and broad interest of our study to the chemistry community.

As Reviewer 1 pointed out, there is only a handful of such challenging studies worldwide that combines NAP-XPS and NAP-STM to monitor the atomic structure and chemical composition of a surface “at work”. Such an approach can be applied not only to catalytic systems, but any functional materials reacting with ambient atmosphere (for instance, sensors) where the surface chemistry plays a decisive role in their applications. Also alloying/dealloying phenomena, especially at surfaces, as found here for Ga-Cu is important for solid state physics. Ultrathin (“monolayer”) oxide films formed or grown on metals is currently a rapidly growing field in catalysis, nano-electronics, sensors, etc. Such two-dimensional films often possess structures having no bulk analogs and may therefore show properties different from those of the bulk counterparts. Therefore, we believe that our study will be of interest to a broad scientific community.

2-1. Lines 39-40: *What do the authors mean by "excellent performance"? Why is the Ni-Ga catalyst superior with respect to other catalysts reported in the literature for CO₂ hydrogenation?*

Authors' response: There is a misunderstanding here. The key word in that sentence that the reviewer might have missed is the reaction pressure, i.e., the NiGa catalysts showed reactivity at atmospheric pressure, with similar or higher methanol yield than the famous Cu/ZnO/Al₂O₃

methanol synthesis catalyst under the same reaction conditions. We admit however that “excellent” sounds too strong, therefore we replaced it by “good”. The revised text reads:

Page 2: *“Recently, it has been discovered that Ni-Ga intermetallic catalysts show good catalytic performance even at atmospheric pressures, with similar or higher methanol yield than the Cu/ZnO/Al₂O₃ methanol synthesis catalyst under the same reaction conditions”.*

2-2. Line 50: *What do the authors mean by "realistic pressure conditions"? CO₂ hydrogenation processes are typically done at high pressures. Can NAP-XPS and NAP-STM be operated under high pressures of CO₂ or H₂?*

Authors’ response: In the surface science community, the term “ambient pressure” or “near ambient pressure” (AP-, NAP-) in conjunction with a particular surface-sensitive technique is used to highlight the fact that such experiments are carried out at pressures up to the atmospheric pressures (typically, 1 – 100 mbar), i.e., many orders of magnitude higher than usually used in “surface science” studies. The reviewer is correct that there is still a pressure gap that in fact prompted us to include in our study also quasi-in situ measurements carried out in a high pressure cell (at 5 bar) directly attached to our analysis UHV system. These measurements were also used as reference in our discussion. Nonetheless, since the acronym NAP present in this sentence already indicates the range of pressures considered, we revised this sentence as follows:

Page 2: *In this work, we took advantage of Near Ambient Pressure X-ray Photoelectron Spectroscopy (NAP-XPS) and Scanning Tunneling Microscopy (NAP-STM), ~~the methods developed for studying surfaces under realistic pressure conditions~~¹⁹⁻²⁷, to monitor ...”*

The methanol synthesis is, indeed, performed at pressures as high as 50 bar that remain unachievable for modern STM and XPS apparatus. Although the NAP pressures are still far below the real reaction pressures, the catalyst surface under NAP conditions often undergoes the same type of structural reconstructions that occur under catalytically relevant pressures. This was demonstrated in our work by comparison of XPS results obtained after reaction at 1 mbar and at 5 bar.

2-3. Lines 60-80: *What is the link between the Ga-Cu(111) model and a real GaCu catalyst? I agree that is very important to study the behavior of Ga-Cu(111), but how far can one extrapolate the results obtained for this surface?*

Authors’ response: This is one of the most important questions for all model studies, which is commonly referred to as the “materials gap” between the real and the model catalysts. To date, the surface structures observed here on Ga/Cu(111) could not be resolved on highly dispersed powder catalysts with the standard characterization techniques applied in catalytic studies. However, we expect that our demonstration of their formation in our planar systems, even under very mild reaction conditions, serves as inspiration for skilled synthetic chemists in the preparation of novel Cu-Ga nano-particulate powder catalysts. Such pre-designed

materials are still likely to further evolve under reaction conditions but might face a more energetically favorable pathway towards the formation of the active state under reaction conditions.

2-4. Lines 60-80: What is the miscibility of Ga and Cu? In the systems shown in Figure 1, are we seeing a surface alloy or a bulk alloy?

Authors' response: Based on the Ga-Cu bulk phase diagram (see, for instance, Li et al, “A thermodynamic assessment of the copper–gallium system”, Calphad 32 (2008) 447) Ga readily intermixes with Cu. As XPS using a laboratory x-ray source provides information about elemental composition of the Ga-Cu surface only about 1 nm in depth (https://hbcpc.chemnetbase.com/faces/documents/12_39/12_39_0001.xhtml), the formation of a real bulk alloy cannot be addressed with this technique. Additional XPS measurements that we have performed at grazing electron emission suggest that Ga is mostly located at the surface and hence forms a surface alloy. We have added this new information as Supplementary Figure 2c and new text addressing this point on page 3:

Page 3: “The fact that the Ga:Cu ratio in the annealed samples considerably increases at grazing electron emission (Supplementary Fig. 2c) suggests that Ga is mostly located in the surface layers and forms a surface alloy.”

Supplementary Figure 2. (c) Ga 2p : Cu 2p signal ratio as a function of the electron emission angle (see inset). The relative increase of the Ga/Cu ratio at grazing emission suggests that Ga is mostly located in the surface layer.

2-5. Figure 2, it would be nice to see sequential STM images for a pristine Ga-Cu(111) surface, a surface exposed to CO₂, and a surface exposed to CO₂ + H₂. Same thing for the NAP-XPS results. It is important to establish what CO₂ alone can do to the Ga-Cu(111) surface.

Authors' response: We thank the reviewer for this suggestion. With respect to the question about what CO₂ alone does to the Ga-Cu surface, we refer to the results of our *quasi in situ* XPS measurements presented in Supplementary Figures 16 and 17 (re-numbered after revision). In pure CO₂, the entire oxidation of Ga and only surface oxidation of Cu(111) takes place, indicating that CO₂ behaves as an oxidizing agent. Subsequent exposure to H₂ (or direct exposure to the CO₂ + H₂ mixture) reduces the Cu-oxide layer, while Ga remains oxidized. We have not done yet NAP-STM measurements in pure CO₂, but we anticipate the formation of the Ga-oxide islands in the same manner as we have observed in the CO₂ + H₂ reaction mixture and also in pure O₂ as shown in Fig. 4. Note also that our primary goal in this study was to monitor structural changes induced by the reaction atmosphere, i.e., in the mixture of CO₂ and H₂.

2-6. *Lines 142-158: The best way to study the possible formation and disappearance of the mentioned hydrocarbon intermediates is by measuring the C 1s XPS or C K-edge NEXAFS.*

2-7. *Lines 295-306: Why the authors did not measure C 1s XPS spectra? Neither Ga nor Cu have XPS features in the 280-290 eV region.*

Authors' response: We measured the C 1s spectra, but the signal to noise ratio was insufficient to draw solid conclusions. As proposed by Reviewer 1, this is most likely due to using a laboratory monochromatic Al K α source. In this respect, using synchrotron sources would be helpful in our future studies, which will be focused now on reaction intermediates and the reaction mechanism on the Ga-promoted Cu surfaces. The latter is outside the scope of the present study. Nonetheless, we have now included the C 1s spectra in the revision as Supplementary Figure 15:

Supplementary Figure 15 | The C 1s region in the NAP-XPS spectra measured in 1 mbar of the $\text{CO}_2 + \text{H}_2$ (1:3) reaction mixture on pure Cu(111) (a) and Ga-Cu(111) (b) surfaces at different temperatures increased stepwise as indicated (from bottom to top). The corresponding O 1s spectra are displayed in Supplementary Figure 7 and Figure 3b in the main text, for Cu and Ga-Cu surfaces, respectively. The signal at 293.2 eV originates from CO_2 in the gas phase. The peak at 284.6 eV is assigned to adventitious carbon.⁽¹⁾

- (1) Deng X, Verdaguer A, Herranz T, Weis C, Bluhm H, Salmeron M. Surface Chemistry of Cu in the Presence of CO_2 and H_2O . *Langmuir* **24**, 9474-9478 (2008).

Reviewer 3

Transition-metal based catalysts, for instance, ZnO-Cu, CeO_x -Cu, have been widely investigated in the field of the heterogeneous catalysis towards CO_2 reduction reaction (CO_2RR). Especially, Gallium-containing alloys are recently demonstrated to be able to hydrogenate CO_2 to methanol at mild conditions. In this context, this work unravels surface structures of the Ga-promoted Cu catalyst during CO_2 hydrogenation, and demonstrates the formation of ultrathin Ga-oxide layer at the metal surface under CO_2 hydrogenation reaction conditions, while the Cu substrate is always kept metallic during reaction. In addition, authors also claim that the Ga-oxide islands are embedded into the copper substrate. Indeed, such kind of structure information at surface/interface is relatively difficult to obtain by using the

conventional techniques as commonly utilized for characterization of powder catalysts.

Although the formation of ultrathin Ga₂O₃ layer embedded in the substrate has not been frequently observed before, it is generally recognized that the morphology of oxide overlayers on metals is a result of a delicate balance between the surface and interface energies. Furthermore, the atomic insight into the Ga-oxide layer is still missing in this work. There are numerous measurements including NAP-STM and NAP-XPS measurements presented in the supporting information, which indeed took considerable efforts to perform, however, the highlight of this work is not that appealing for readers. From both the weakness of novelty as similar reports found in literature, and the lack of fundamental mechanism investigation either in the reaction path/products selectivity or in the origin of metallic behavior of the Cu substrate during reactions. Thus, I come to the conclusion with regret that draft under its current version is not convincing enough to be published in Nature communications, but more suitable for the specialized journal, for example, Journal of catalysis or Journal of physical chemistry letters. Meanwhile, there are also a few concerns needed to be addressed:

Authors' response: We thank the reviewer for the time evaluating our work in detail and this important general comment, whose critical aspects we would like to address separately:

- 1) *“Although the formation of ultrathin Ga₂O₃ layer embedded in the substrate has not been frequently observed before, it is generally recognized that the morphology of oxide overlayers on metals is a result of a delicate balance between the surface and interface energies”*

Authors' response: We would like to clarify here that, to the best of our knowledge, the formation of embedded Ga-oxide layers into the metal surface has never been observed, certainly not for the Cu-Ga system, and that it is problematic to extrapolate the behavior of a catalytic material system based on data from different metal-oxide combinations, and thus, experimental evidence such as the one provided here describing *in situ* the evolution of the structure and composition of the catalyst in a reaction environment is essential for mechanistic understanding of the catalytic process. Moreover, as also mentioned by Reviewer 1, there is only a handful of such challenging studies worldwide (e.g. combination of NAP-XPS and NAP-STM, and here in addition quasi *in situ* high pressure experiments) and having access to such experimental data is the key to the understanding of complex catalytic interfaces while at work.

- 2) *“Furthermore, the atomic insight into the Ga-oxide layer is still missing in this work.”*

Authors' response: Ultrathin (“monolayer”) oxide films formed or grown on metals is currently a rapidly growing field in catalysis, nano-electronics, sensors, etc. Such films often possess structures having no bulk analogs, therefore, in-depth understanding of their atomic structures is not a simple task and requires substantial additional work in combination with theoretical calculations. Indeed, in the examples cited in the manuscript, it took years before

the atomic structures were ultimately established. Our combined XPS and STM results provide just the first evidence for the formation of a “monolayer” Ga-oxide on a metal substrate (in this case, Cu) under CO₂ hydrogenation reaction conditions. Further studies on its atomic structure will be the subject of future investigations, but its challenge under NAP conditions cannot be underestimated and should not hold the publication of the present work.

3) “...*the weakness of novelty as similar reports found in literature...*”

Authors’ response: We respectfully disagree that our work lacks novelty. To the best of our knowledge, there are no publications reporting a combination of NAP-STM and NAP-XPS for studying Ga-containing model catalysts, in particular in the CO₂ hydrogenation reaction where the formation of Ga oxide in a H₂-rich atmosphere is not obvious. Using NAP-STM at high temperatures, we showed *in situ* with sub-nanometer atomic resolution de-alloying of the Ga/Cu(111) surface in the reaction atmosphere. The formation of a Ga-oxide ultrathin layer (“monolayer”) embedded into the metal surface has not been reported, either.

4) “...*the lack of fundamental mechanism investigation either in the reaction path/products selectivity or in the origin of metallic behavior of the Cu substrate during reactions.*”

Authors’ response: We note that this work is, first of all, about surface structure(s) formed under reaction conditions monitored *in situ* by two complementary surface-sensitive techniques. Such kind of structural information is a prerequisite for elucidating the reaction mechanism and is largely missing in the literature. This insight is however of outmost importance for theoreticians calculating reaction pathways, since we have now provided the required information on the structure of the surface under reaction conditions. Theorists can now use this work as starting point for their mechanistic study instead of the pristine Cu-Ga metallic surface that we know now is not stable during reaction. In addition, we could also shed some light on the reaction mechanism experimentally by comparison of the NAP-XPS results on pure Cu(111) and Ga/Cu(111) surfaces. On Cu(111), only carboxylate was found at 300 K that stays at 373 K and desorbs at 473K without any reaction with H₂ (Supplementary Figure 7). In contrast, on Ga/Cu(111), additional O 1s states at 532.2 and 531.0 eV appear at elevated temperatures (373 K and above). The former can be assigned to formate based on the previous XPS studies reported in the literature and IR studies on Ga₂O₃ powders. As we wrote in the text, “... Their formation would suggest H₂ dissociation on the Cu(111) surface surrounding the Ga-oxide islands, and subsequent H spillover onto the oxide surface. The hydrogen dissociation is also a precondition for the Ga-oxide reduction to metallic Ga (Fig. 3b). ...” The origin of another state at 531.0 eV is still unclear. We discuss some hypothesis in Supplementary Note 2 in SI.

As to the metallic behavior of Cu during the reaction, this has previously been shown for several Cu-based catalysts elsewhere (e.g., refs. 31, 32) and is commonly associated with the hydrogen-rich atmosphere that is used for methanol synthesis reaction (CO₂ : H₂ = 1:3).

3-1. *The ratio of CO₂:H₂ was chosen to be 1:3 in this work, is there any concern about this? Has authors tried different partial ratio of pressure, say, 1:2 or 1:4 to see if any difference?*

Authors' response: We did not vary the CO₂ : H₂ ratio in the reaction mixture. This ratio (1:3) reflects stoichiometric equation for methanol synthesis (CO₂ + 3H₂ → CH₃OH + H₂O) and was chosen since it is commonly used in related literature studies. We have now added this information in experimental section:

Page 10: *“In all experiments using the reaction mixture, we used the ratio CO₂ : H₂ = 1 : 3 that reflects a stoichiometric equation for the methanol synthesis reaction (CO₂ + 3H₂ → CH₃OH + H₂O) and has been commonly used in related prior studies.”*

3-2. *'To complete the STM and XPS correlation, we also present the spectra obtained on an identical but newly prepared sample after a NAP-STM study at 1 mbar in the same reaction mixture at 300 K...' in page 4, related spectra is not shown or clearly mentioned in the draft.*

Authors' response: We apologize for this confusion. In the revised text, we now refer to the spectra labelled (ii) in Fig. 2h:

Page 4: *“To complete the STM and XPS correlation, we also present the spectra obtained on an identical but newly prepared sample after a NAP-STM study at 1 mbar in the same reaction mixture at 300 K (see (ii) in Fig. 2h).”*

3-3. *Figure 2c is labelled two times by mistake in Figure 2.*

Authors' response: We thank the reviewer for careful reading of the manuscript. We eliminated the typo in Fig. 2b.

3-4. *The deconvolution of O1s spectra with two peaks at 531.0 and 531.4 eV (binding energy), is not convincing since the difference is such tiny, especially in the case of the spectrum at 300 K in Figure 3b where the fitting is not proper. Thus, some parts of discussion might be revised accordingly.*

Authors' response: Spectral deconvolution is always a non-trivial procedure. In our analysis of the O 1s spectra, we used two firm references, namely: (i) the 530.2 eV peak assigned to the lattice oxygen in the Ga-oxide layer (proven by inspection of the sample treated with molecular oxygen, see Suppl. Fig. 20); and (ii) the 531.4 eV peak assigned to carboxylate based on experiments performed on a pure Cu sample (Suppl. Fig. 7), also well-documented in the literature. Having this at hand, we could only fit the spectra obtained at elevated temperatures by invoking the additional state centered at 531.0 eV. Although it is very close to carboxylate

at 531.4 eV, the difference is considerably larger than our experimental error margin with our monochromatic AlK α source. Note, however that this state was only detected at temperatures of 473 K and above, where the signal of lattice oxygen at 530.2 eV noticeably decreases due to the partial reduction of the Ga-oxide revealed by the Ga 2p spectra shown in Fig. 3b. This new state is not obvious and therefore was not included in the deconvolution of the spectra at 300 and 373 K. The origin of the new state is not fully understood. We speculate on this matter in Suppl. Note 2. Nonetheless, following the reviewer’s comment, we slightly improved deconvolution in Fig. 3b for the spectrum at 300 K as shown below:

Figure 3. (revised)